# Exploring the Role of Gut Microbiota in Patients with Alopecia Areata

**DOI:** 10.3390/ijms25084256

**Published:** 2024-04-11

**Authors:** Ji Hae Lee, Ji Hae Shin, Ji Yoon Kim, Hyun Jeong Ju, Gyong Moon Kim

**Affiliations:** Department of Dermatology, St. Vincent’s Hospital, College of Medicine, The Catholic University of Korea, Seoul 16247, Republic of Korea; shinjihae1001@gmail.com (J.H.S.); kjy.jenny@gmail.com (J.Y.K.); hyd0116@naver.com (H.J.J.); gyongmoonkim@catholic.ac.kr (G.M.K.)

**Keywords:** alopecia areata, gut microbiome, dysbiosis

## Abstract

Imbalances in gut microbiota reportedly contribute to the development of autoimmune diseases, but the association between the etiopathogenesis of alopecia areata (AA) and gut microbial dysbiosis remains unclear. This cross-sectional study was conducted to identify and compare the composition of the gut microbiome in patients affected by AA and those in a healthy control (HC) group, and to investigate possible bacterial biomarkers for the disease. Fecal samples were collected from 19 AA patients and 20 HCs to analyze the relationship with fecal bacteria. The three major genera constituting the gut microbiome of AA patients were *Bacteroides*, *Blautia*, and *Faecalibacterium*. The alpha diversity of the AA group was not statistically significant different from that of the HC group. However, bacterial community composition in the AA group was significantly different from that of HC group according to Jensen–Shannon dissimilarities. In patients with AA, we found an enriched presence of the genera *Blautia* and *Eubacterium_g5* compared to the HC group (*p* < 0.05), whereas *Bacteroides* were less prevalent (*p* < 0.05). The gut microbiota of AA patients was distinct from those of the HC group. Our findings suggest a possible involvement of gut microbiota in in the as-yet-undefined pathogenesis of AA.

## 1. Introduction

Alopecia areata (AA) is a non-scarring hair-loss disease characterized by a T-lymphocyte-mediated autoimmune response to anagen hair follicles. AA occurs in 0.1% to 0.2% of the population, with a lifetime risk of approximately 1.7% to 2.1% [1,2]. Typically, AA presents as well-demarcated patches of hair loss on the scalp. It typically follows an unpredictable clinical course, often recurring or progressing to involve the entire body. AA negatively impacts self-esteem and body image, consequently impairing social skills and diminishing quality of life [3]. The pathogenesis of AA remains incompletely understood; the most widely accepted theory involves complex interactions between genetic or epigenetic predispositions and unspecified triggering factors, leading to a local surge of interferon-gamma that collapses the hair-follicle-immune privilege and elicits an autoimmune response targeting exposed hair follicle autoantigens by activating autoreactive cytotoxic T cells [4]. Recent advances in immunology have revealed that an imbalanced ratio of Th17 lymphocytes and regulatory T (Treg) cells is involved in the pathophysiology of autoimmune processes, and increased levels of Th17 cells and decreased levels of Treg cells have been found in AA patients [5].

Recent advancements in genome sequencing have revealed the significant influence of gut microbiota in various autoimmune diseases, highlighting the role of gut microbiota in modulating immune responses [6]. Normal gut microbiota is essential to the development and maturation of the intestinal mucosal immune system, and specific microbial groups induce differentiation and activation of immune cells, regulating the balance between immune tolerance and immune stimulation. An imbalance in the intestinal microbiome can weaken intestinal defense barriers and damage intestinal mucosa. Ultimately, pathogens, toxins, and antigens in the intestinal tract enter the bloodstream and stimulate the immune system, potentially leading to autoimmune diseases, including inflammatory bowel disease (IBD), systemic lupus erythematosus, and Hashimoto’s thyroiditis [7,8,9]. Gut microbiota also reportedly play a crucial role in regulating the systemic balance between Th17 cells and Treg cells [10]. Short-chain fatty acids synthesized by gut microbiota modulate the activity of Treg cells, which are critical to maintaining immune tolerance and potentially to development of AA [11].

Several independent lines of evidence support the idea that gut microbiota can contribute to the pathogenesis of AA. Rebello and Xie observed hair regrowth in AA patients following fecal material transplantation for refractory bowel diseases [12,13]. Preclinical studies in murine models have indicated the possible role of diet and the gut microbiome in the onset of AA [14,15]. Recent clinical studies have characterized the gut microbiota of AA patients [16,17,18,19]; however, the role of microbiota in AA remains unclear. This knowledge gap presents a significant opportunity for research that elucidates the potential link between gut microbiota and AA. This study aimed to analyze the characteristics of the gut microbiomes in AA patients compared to healthy controls (HCs), and, subsequently, to identify effective bacterial biomarkers for the disease and potential therapeutic targets. By understanding the specific alterations in the gut microbiome associated with AA, this study seeks to obtain novel insights into the disease’s pathogenesis and explore new avenues for treatment, potentially paving the way for microbiome-targeted therapies.

## 2. Results

### 2.1. Clinical Characteristics

We recruited 19 patients (8 females and 11 males) diagnosed with AA for the study. The mean age of the AA group was 44.6 ± 14.1 years; 2 patients were taking medication for hypertension, and 4 were current smokers. The mean duration of AA was 12.0 months, and 15 patients experienced AA for the first time. Among these patients, 16 had patchy AA, and 3 had diffuse AA. The severity of. AA was assessed using the Severity of Alopecia Tool (SALT) score, with 11 patients classified in the limited AA group as SALT grade S1, and 8 patients in the moderate-to-severe AA group ranging from SALT grades S2 to S5. The HC group consisted of 20 individuals (14 males and 6 females) with an average age of 50.5 ± 4.2 years.

### 2.2. Gut Microbial Composition and Diversity in AA Patients and Controls

We obtained an average of 47,262 valid reads per sample (range: 26,417–85,273) and identified an average of 3441 chimeric amplicons per sample (range: 450–12,568). The average read coverage achieved across all samples was 96.27%, with an average of 310 species identified per sample.

The major genera that constitute the gut microbiome’s core were *Bacteroides*, *Blautia*, *Faecalibacterium,* and *Prevotella* in both AA and HC groups (Figure 1). In the AA group, the most observed genera were in the order of *Bacteroides*, *Blautia*, *Faecalibacterium*, and *Prevotella*, while in the HC group, they were observed in the order of *Bacteroides*, *Prevotella*, *Faecalibacterium*, and *Megamonas.* A distinct difference in the proportions occupied by the microbiota was observed. The *Firmicutes* to *Bacteriodetes* (F/B) ratio was significantly higher in the AA group compared with the HC group (*p* < 0.05; Figure 2).

The analysis of alpha diversity revealed that bacterial richness (ACE and Chao1) and evenness (Shannon index and Simpson function) of the AA group were not significantly different from those of the HC group (*p* > 0.05; Figure 3). The mean ± standard deviation (SD) of the alpha diversity, assessed using the Shannon index, in the AA group was 3.46 ± 0.534.

Principal coordinate analysis results were plotted based on unweighted UniFrac dissimilarity between samples to compare the compositions of the gut microbial communities. The bacterial community compositions in AA patients were significantly different from those of HCs according to Jensen–Shannon dissimilarities (*p* < 0.05; Figure 4).

We searched for bacterial biomarkers for AA using linear discriminant analysis (LDA) effect size (LEfSe). In patients with AA, we found an enriched presence (LDA score > 3) of genera *Blautia*, *Dorea*, *Collinsella*, *Anaerostipes,* and *Eubacterium_g5* compared with the HC group (*p* < 0.05; Figure 5). In contrast, fewer members of the family *Ruminococcaceae* and species *Bacteroides* were present in patients with AA (*p* < 0.05; Figure 5).

### 2.3. Gut Microbial Composition and Diversity in Patients with Limited AA and Moderate to Severe AA

The patients with AA were divided into two groups according to SALT score. The limited AA group consisted of patients with an SALT grade of S1. The moderate-to-severe AA group consisted of patients with an SALT grade of S2 to S5. The two groups showed no significant difference in alpha diversity (*p* > 0.05; Figure 6) and beta diversity. The limited AA group showed enriched presence of *Coprococcus* and *Saccharimonas* (*p* < 0.05).

## 3. Discussion

The microbiome exists in various parts of the human body, but the gastrointestinal tract contains the largest number and variety of microorganisms. It is estimated that the weight of microorganisms in the gastrointestinal tract is approximately 0.5 to 1.5 kg, and 500 to 1000 species of bacteria live there [20]. Human intestinal microorganisms have a diverse community structure for each individual depending on genetics, eating habits, and lifestyle habits from birth [21]. The collection of genes possessed by these microbial communities is defined as the gut microbiome. Research has demonstrated that the gut microbiota plays an important role in nutrient absorption, drug metabolism, immune system regulation, brain/behavioral development, and prevention of infectious diseases. The gut microbiota contains more than 150 times more genes than the human genome and produces a wide range of enzymes that the human body cannot make [6]. It not only decomposes and absorbs complex carbohydrates and fiber, but produces short-chain fatty acids (SCFAs) such as acetate, propionate, and butyrate in the intestine. These fat metabolites play an important role in regulating the immune system, producing vitamins, and protecting the intestinal mucosa from pathogens [22].

Several studies have identified a potential role for gut microbiota in the pathogenesis of AA. In murine models, normal-haired C3H/HeJ mice developed AA after receiving skin grafts from mice affected by AA, but the proportion of AA development decreased when the mice were fed with a diet enriched with soy oil [14]. In the same animal model, treatment with broad-spectrum antibiotics prevented the development of AA and was associated with reduced T cell infiltration in the skin [15]. Cases of hair regrowth after fecal transplantation in patients with intractable intestinal disease accompanied by AA have been reported [12,13], and recent clinical studies have characterized the gut microbiota of patients with AA. A cross-sectional study by Moreno-Arrones et al. compared the gut microbiome of patients with alopecia universalis with those of controls by sequencing the 16SrRNA extracted from stool samples [18]. Although no significant differences in α and β diversity were found between the two groups, this study developed a predictive model with a diagnostic efficacy of 0.8, based on the abundance of two bacteria: *Parabacteroides distasonis* and the *Clostridiales vadin* BB60 group [18]. A study by Rangu et al. investigated the gut microbiota in pediatric AA patients using shotgun metagenomics [16]. While the composition and diversity of the microbiota in AA patients and their siblings were similar, gene functionality analysis revealed differences in 20 genes [16]. Lu et al. compared the gut microbiota of AA patients and healthy controls from China [17]. There were no significant differences in α diversity observed between the two study groups; however, they adopted a random forest model to select three AA-associated biomarkers, such as *Achromobacter*, *Megasphaera,* and *Lachnospiraceae incertae sedis*. The study by Brzychcy et al. analyzed stool samples from 25 patients with active AA. They identified a microbiome characterized by an overrepresentation of *Firmicutes* and *Proteobacteria*, alongside a decrease in overall richness and taxonomic diversity [19]. The findings of clinical studies on the gut microbiome in AA can be influenced by the diversity of the study population (e.g., patients with alopecia universalis and children with AA), geographical location, and lifestyle. Therefore, caution is needed in interpretation.

Building on this background, we investigated the involvement of gut microbiota in AA by examining the differences in gut microbiota between AA patients and an HC group. Our findings reveal that, while the overall diversity of microbial composition did not differ significantly, disparities were evident in the relative abundance and distribution of gut microbiota between the AA and HC groups. This aligns with the finding of a study by Lu et al., who reported a similar degree of alpha diversity but significant differences in beta diversity between AA and HC groups [17]. However, neither Moreno-Arrones et al. nor Rangu et al. found significant differences in alpha and beta diversities among their respective study groups [16,18]. Their results may have been affected by the fact that the study subjects were alopecia universalis patients and children with AA.

Our study found a significant enrichment of *Blautia*, *Collinsella*, and *Dorea* in AA patients, suggesting a potential role of the genera in the pathophysiology of AA. An increased presence of *Blautia* has been associated with autoimmune diseases, including Hashimoto’s thyroiditis, psoriasis, and rheumatoid arthritis [9,23]. Similarly, the genera *Collinsella* and *Dorea* were found at elevated levels in patients with IBD, chronic rheumatic diseases, autoimmune thyroid disease, Parkinson’s disease, and psoriasis [9,24,25]. The specific role of these genera in various autoimmune diseases requires further investigation.

We also observed a reduced level of *Bacteriodes* in the AA group, consistent with a report by Moreno-Arrones et al. [18]. *Bacteriodes* is a Gram-negative group of bacteria that constitute a significant portion of human microbiota [26]. They have a symbiotic host–bacterial relationship with humans, helping with the digestion of food and production of nutrients [26]. *Bacteriodes* also produce molecules such as polysaccharide A and Toll-like receptor 2, which are known to activate T-cell-dependent immune responses. A lower abundance of *Bacteriodes* was seen in patients with IBD, and AA was more prevalent in patients with IBD [27]. There may be a shared dysregulation of immune function in both diseases, and a lower abundance of *Bacteroides* may affect pathogenesis [28].

The F/B ratio was higher in the AA patients compared to the HCs, suggesting bacterial dysbiosis. *Firmicutes* and *Bacteriodetes* are two major phyla of human gut microbiota [29]. The F/B ratio is used as a measure of dysbiosis, as it reflects the balance of intestinal symbiotic microbiota. Several studies have shown that the F/B ratio in psoriasis patients is higher compared with that in a control group [24,30]. A positive correlation with Psoriasis Area Severity Index score has also been reported [31]. The elevated F/B ratio observed in our study among AA patients further suggests that dysbiosis may play a role in the pathogenesis of AA, underscoring the potential for microbial balance as a target for therapeutic intervention. The F/B ratio is affected by various factors such as obesity and diet, and the usefulness of the F/B ratio in autoimmune diseases requires further research [29].

Short-chain fatty acids, which are metabolized by gut microbes from soluble fibers and oligosaccharides, serve as an energy source for gut microbiota and intestinal cells [32]. They also regulate T cell gene expression and promote regulatory T cell proliferation. The *Ruminococcoceae* family is a major bacterium that produces butyrate, which is a type of SCFA [33]. Butyrate plays a role in the function and proliferation of Tregs and maintenance of intestinal barriers [34]. The decrease in *Ruminococcoceae* seen in AA patients may contribute to its pathogenesis by lowering the level of SCFAs and disintegrating the intestinal barrier and immune regulation. However, various bacteria contribute to the production of SCFAs, and a combined effect should be considered.

Regarding the alpha and beta diversity in cases of limited and severe AA, no significant differences were observed, suggesting a clustered microbiota composition within the AA group. This finding is consistent with Lu et al.’s study, which also reported no significant diversity differences between mild and severe AA groups [17]. However, the number of patients in each subgroup was too small for a robust analysis of differences according to severity, necessitating caution in the interpretation of these results.

This study is subject to several limitations, including a relatively small sample size and the collection of data from a single center. Additionally, there was inadequate matching for age and body mass index among the cohorts, and the diets of the participants were not controlled. To more conclusively determine the impact of gut microbiota on the pathogenesis of AA, further studies with larger sample sizes and incorporate controlled dietary factors are needed.

## 4. Materials and Methods

### 4.1. Participants

This study was conducted at St. Vincent’s Hospital, Suwon, Republic of Korea. Patients with AA were recruited from the hospital’s outpatient clinics between May 2021 and October 2022. Inclusion criteria were age > 18 years and clinically proven AA. The diagnosis of AA was based on typical clinical and trichoscopic manifestations. Exclusion criteria were co-existing scalp dermatoses or other hair loss conditions, history of receiving oral or intravenous antibiotic treatment, systemic immunomodulators, extreme diets, or probiotics in the previous 12 weeks. Age- and sex-matched HC data were extracted at random from a cohort of men and women in Korea using the EZBioCloud public database [35,36]. The study was performed in accordance with the principle of the Declaration of Helsinki. The Institutional Review Board at Catholic Medical Center reviewed and approved the protocol. Written informed consent for microbiota analysis and the use of personal data for research purposes was obtained from subjects before treatment.

### 4.2. *Stool Sample Collection* and DNA Extraction

Fecal samples were collected using a fecal collection kit (SMF-1; ChunLab Inc., Seoul, Republic of Korea) that contained a lysis buffer (SDS 4%, Tris–HCL 50 mM, EDTA 50 mM, NaCl 500 mM) [37,38]. Patients were instructed to take a sample of their first feces in the morning and place it in the kit. Samples were stored frozen until arrival at the laboratory, which took less than 3 days.

The samples were frozen at −80 °C for 16S RNA gene sequencing and DNA extraction. Total DNA was extracted using a Maxwell RS Purefood GMO and Authentication Kit (Promega, Madison, WI, USA), in accordance with the manufacturer’s instructions. The region from V3 to V4 of 16S rRNA gene was targeted and amplified with fusion primers 341F and 805R. The amplified polymerase chain reaction products were verified using 1% agarose gel and visualized under a Gel Doc system (BioRad, Hercules, CA, USA). CleanPCR (CleanNA, Waddinxveen, The Netherlands) was used to purify the products. The quality and size of the purified products were assessed with a Bioanalyzer 2100 (Agilent, Palo Alto, CA, USA) using a DNA 7500 chip. Mixed amplicons were sequenced at CJ Biosicence, Inc. (Seoul, Republic of Korea) according to an Illumina MiSeq Sequencing system (Illumina, San Diego, CA, USA).

### 4.3. Statistical and Bioinformatic Analysis

The microbiome taxonomic profiling cloud of EZBiocloud was used to analyze the taxonomic bacterial profiling [39]. After normalization based on 16S rRNA gene copy number variation, various analyses were conducted: alpha- and beta-diversity indices and LEfSe for identification of biomarkers [40,41]. Species richness was measured by Chao, ACE, and number of operational taxonomic units. Diversity indices were expressed with Shannon and Simpson indices. Beta diversity was measured using Bray–Curtis distances, and the statistical significance was analyzed using permutational multivariate analysis of variance. For LEfSe analysis, taxonomic levels with an LDA score higher than 3 and *p*-value < 0.05 were considered statistically significant [18,40].

## 5. Conclusions

While the richness and evenness of the gut microbiota in the AA group did not differ significantly from those in the HC group, notable distinctions were observed in the relative abundance and composition of bacteria between the two groups. Specifically, the enrichment of certain bacteria in the AA group mirrors patterns seen in other autoimmune diseases, suggesting a link between gut microbiota and the pathophysiology of AA. However, whether these microbial differences were a cause or a consequence of the disease is unclear. This uncertainty underscores the need for further, more detailed research to elucidate the exact role of gut microbiota in autoimmune skin disorders. Such studies are vital not only to deepen our understanding of these conditions but also to explore the potential of gut microbiota as biomarkers and therapeutic targets in AA and related diseases.

## Figures and Tables

**Figure 1 ijms-25-04256-f001:**
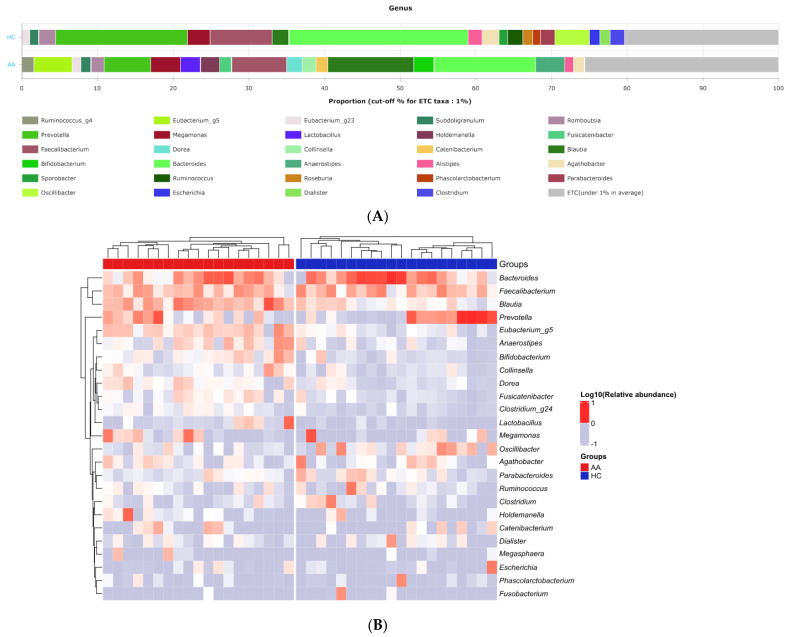
The gut microbial composition of the alopecia areata and health control groups. (**A**) Averaged taxonomic compositions at the genus level. (**B**) Heatmap of taxonomic compositions at the genus level. AA, alopecia areata; ETC, etcetera; HC, healthy control.

**Figure 2 ijms-25-04256-f002:**
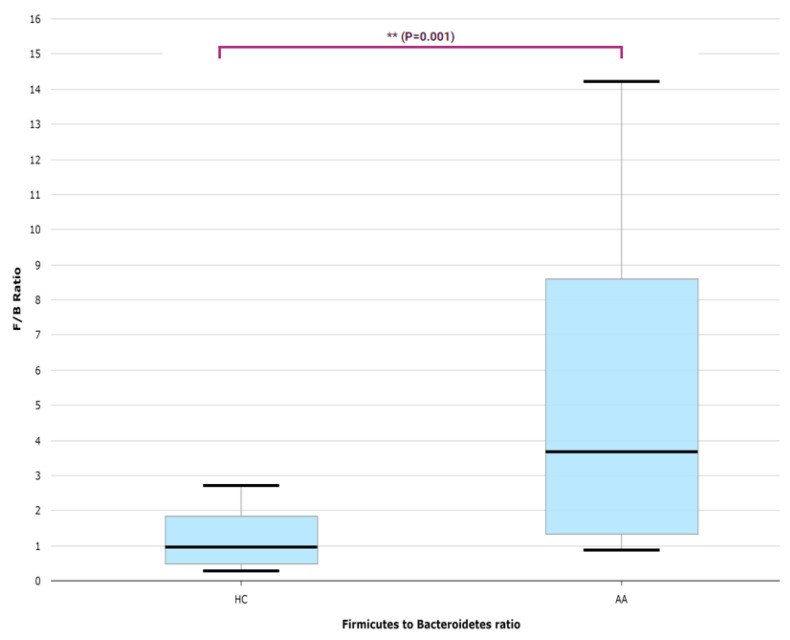
*Firmicutes* to *Bacteriodetes* (F/B) ratio of the alopecia areata and health control groups. AA, alopecia areata; HC, healthy control.

**Figure 3 ijms-25-04256-f003:**
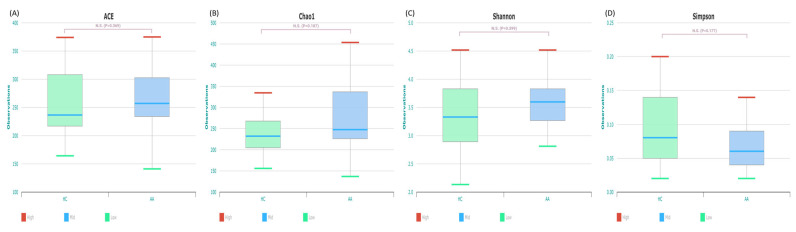
The α diversity of gut microbiota of the alopecia areata and health control groups. There were no statistically significant differences between the two groups with ACE (**A**), Chao1 (**B**), Shannon (**C**), and Simpson (**D**). AA, alopecia areata; HC, healthy control.

**Figure 4 ijms-25-04256-f004:**
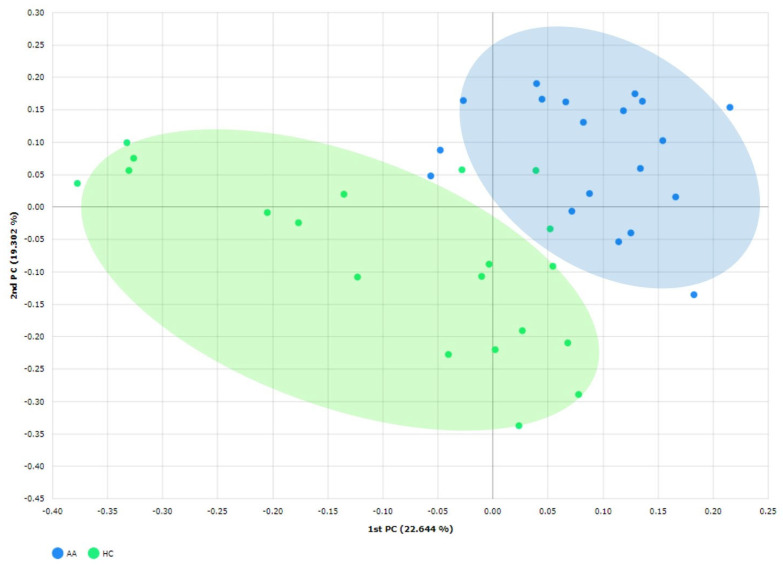
Principal coordinate analysis (PCoA) for the alopecia areata and healthy control groups. PCoA was plotted based on unweighted-UniFrac dissimilarity between samples. The ellipses highlight the clustering of the gut microbiota according to groups. AA, alopecia areata; HC, healthy control.

**Figure 5 ijms-25-04256-f005:**
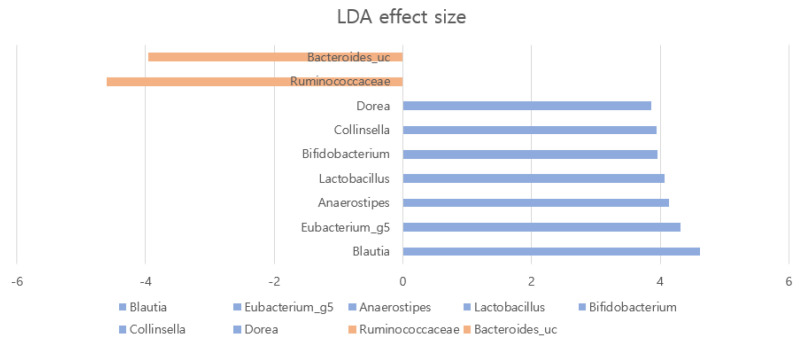
Crucial bacteria of gut microbiota related to alopecia areata. Relative abundance of bacterial taxa calculated by the linear discriminant analysis effect size (LEFse) tool.

**Figure 6 ijms-25-04256-f006:**
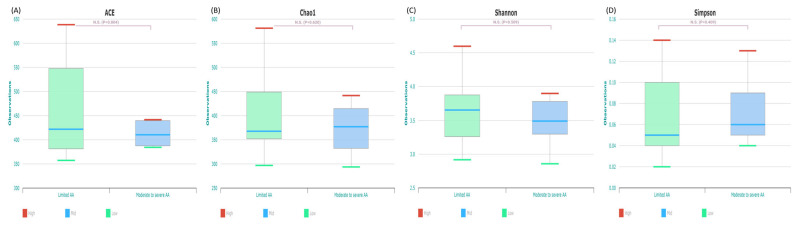
The alpha diversity of gut microbiota of the limited alopecia areata and moderate to severe alopecia areata groups. There were no statistically significant differences between the two groups with ACE (**A**), Chao1 (**B**), Shannon (**C**), and Simpson (**D**). AA, alopecia areata.

## Data Availability

Data are contained within the article.

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
