# Peer review of "Exploring the Role of Gut Microbiota in Patients with Alopecia Areata"

_ijms, 2024, doi:10.3390/ijms25084256_

Round 1

Reviewer 1 Report

Comments and Suggestions for Authors

The aim of this study was to identify and compare the composition of the gut microbiome of patients with alopecia areata (AA) and healthy controls (HC) and to investigate potential bacterial biomarkers associated with the disease. The main findings were: The top three genera of the gut microbiome of AA patients were Bacteroides, Blautia and Faecalibacterium; the bacterial community composition was significantly different between AA and HC groups, with Blautia and Eubacterium_g5 being more abundant and Bacteroides less abundant in AA patients; however, overall alpha diversity was not statistically different. The study suggests a possible involvement of gut microbiota dysbiosis in the pathogenesis of AA, although the underlying mechanisms require further investigation. Although the study is interesting and well designed, there are some questions that still need to be clarified.

Introduction

1- The authors' claim that "Several independent lines of evidence support the idea that gut microbiota can contribute to the pathogenesis of AA" is not adequately supported by references in the introduction. While the introduction does discuss the general role of gut microbiota in modulating immune responses and contributing to autoimmune diseases, it does not provide specific references to studies that have investigated the link between gut microbiota and alopecia areata. The authors should either provide the appropriate references to support their claim or rephrase the statement to acknowledge the lack of direct evidence on the relationship between gut microbiota and AA pathogenesis, as indicated by their subsequent statement that "few studies have characterized the gut microbiota of AA patients, and the role of microbiota in AA remains unclear."

Materials & Methods

Participants

-The authors should provide brief information on the clinical evaluation and diagnostic criteria used to confirm the diagnosis of alopecia areata in the patient group.

- If the healthy control data were extracted from a pre-existing biobank or registry, including details about the source, data collection methods, and any potential biases or limitations associated with this approach would improve the transparency of the study.

DNA extraction and sequencing

- The reference provided (reference 34) does not seem to contain any information about the fecal collection kit (SMF-1) or the manufacturer's instructions. If this fecal collection kit is not a widely used or commercially available product, the authors should consider providing additional details about its provenance, validation, and suitability for gut microbiome studies.

- How did the authors assess the qualities and quantities of extracted DNA.

- Please provide the manufacturer’s name, city, country of the CleanPCR. This also valid for all reagents used.

Statistical and bioinformatic analysis

- The authors are requested to provide more details about the data processing such as the database used to annotate the raw sequences and the threshold of identity percentage

- Why LDA score ≥ 2 was used to estimate the effect size of each taxon?

- I`m not sure if the authors made their sequences publicly available or not? If not, they need to submit their sequences and give their deposit number or doi.

- The authors may need to explore their data and use the Phylogenetic Investigation of Communities by Reconstruction of Unobserved States (PICRUSt) approach to analyze both the relative abundances of the 16S rRNA genes and the predicted metabolic pathway.

Results

- Authors should provide information on the number of reads, chimeric sequences, valid reads, and read coverage.

-The low resolution and lack of clear differentiation between the colors representing the bacterial genera make it difficult to discern the distinct differences of the core microbiota between the AA and HC groups.

-Figure 1. What the ETC stand for?

- Line 74-76: Figure 1 also shows that Prevotella accounts for more than 15% of the gut microbiome composition in the HC group, which is a significant proportion. Additionally, the prevalence of Prevotella in the AA group is almost comparable to that of Faecalibacterium, another genus identified by the authors as a core member. Therefore, I would suggest that the authors consider including Prevotella as an additional major genus, particularly given its substantial representation in both the AA and HC groups.

- The authors should specify that the limited AA group consisted of 11 patients with a SALT grade of S1, and the moderate-to-severe AA group consisted of 8 patients with a SALT grade of S2 to S5.

- Again, the authors should consider increasing the resolution of the figures presented in the manuscript. Higher resolution figures would improve the clarity and visual quality of the data visualization, making it easier for readers to interpret the information conveyed in the images.

Discussion:

- The claim about the number of bacterial species in the human gastrointestinal tract being 500-1,000 originated from the review article by Jian Xu and Jeffrey I. Gordon (PMID 12923294), and not from the reference cited by the authors.

- Line 138-143. The authors should reconcile the apparent contradiction in their statements regarding the existing studies on the role of gut microbiota in the pathogenesis of AA. On one hand, they state that "several studies have identified a potential role for gut microbiota in the pathogenesis of AA", suggesting a substantial body of research on this topic. However, they then go on to say that "studies characterizing the gut microbiota in AA patients have been limited".

- Line 154-161. The authors need to provide potential explanations for the observed differences. The authors acknowledge that the findings of Moreno-Arrones et al. and Rangu et al. may have been affected by the study populations (alopecia universalis patients and children with AA), but they do not explore this aspect in more detail. It would be valuable to discuss potential factors that could contribute to the varying results reported in the literature.

- Line 161 correct “univeralis” to “universalis”

- Lines 162-168. The authors have noted the increased incidence of these genera in AA patients, but they do not provide mechanistic insights into how these specific bacteriomes may contribute to the pathophysiology of AA. This is also true for the F/B ratio and the less dominant family Ruminococcaceae

- Lines 180-181 provide an adequate reference (e.g PMID: 32438689)

- Line 178-185. What is the potential clinical implications of this observation?

- Some editing for English language is required throughout the manuscript

Comments on the Quality of English Language

Some editing for English language is required throughout the manuscript

Author Response

First and foremost, we would like to express our sincere gratitude for your thorough review and valuable comments on our manuscript. Your insights have significantly contributed to enhancing the quality and clarity of our study. We have carefully considered your suggestions and have made the following revisions to our manuscript to address the concerns raised.

Comments and Suggestions for Authors

The aim of this study was to identify and compare the composition of the gut microbiome of patients with alopecia areata (AA) and healthy controls (HC) and to investigate potential bacterial biomarkers associated with the disease. The main findings were: The top three genera of the gut microbiome of AA patients were Bacteroides, Blautia and Faecalibacterium; the bacterial community composition was significantly different between AA and HC groups, with Blautia and Eubacterium_g5 being more abundant and Bacteroides less abundant in AA patients; however, overall alpha diversity was not statistically different. The study suggests a possible involvement of gut microbiota dysbiosis in the pathogenesis of AA, although the underlying mechanisms require further investigation. Although the study is interesting and well designed, there are some questions that still need to be clarified.

Introduction

1- The authors' claim that "Several independent lines of evidence support the idea that gut microbiota can contribute to the pathogenesis of AA" is not adequately supported by references in the introduction. While the introduction does discuss the general role of gut microbiota in modulating immune responses and contributing to autoimmune diseases, it does not provide specific references to studies that have investigated the link between gut microbiota and alopecia areataThe authors should either provide the appropriate references to support their claim or rephrase the statement to acknowledge the lack of direct evidence on the relationship between gut microbiota and AA pathogenesis, as indicated by their subsequent statement that "few studies have characterized the gut microbiota of AA patients, and the role of microbiota in AA remains unclear."

Response:   Thank you for your thorough review and valuable comments on our manuscript. In accordance with your suggestion, we have provided appropriate references to support our claim.

The relevant additions are as follows: Several independent lines of evidence support the idea that gut microbiota can contribute to the pathogenesis of AA. Rebello and Xie observed hair regrowth in AA patients following fecal material transplantation for refractory bowel diseases [12,13]. Preclinical studies in murine models have indicated the possible role of diet and the gut microbiome in the onset of AA [14,15]. Recent clinical studies have characterized the gut microbiota of AA patients [16-18]; however, the role of microbiota in AA remains unclear.

Materials & Methods

Participants

-The authors should provide brief information on the clinical evaluation and diagnostic criteria used to confirm the diagnosis of alopecia areata in the patient group.

Response: Thank you for your comments. In accordance with your suggestion, we have made the following additions to the content: The diagnosis of AA was based on typical clinical and trichoscopic manifestations.

- If the healthy control data were extracted from a pre-existing biobank or registry, including details about the source, data collection methods, and any potential biases or limitations associated with this approach would improve the transparency of the study.

Response: Thank you for your valuable comments. We would like to provide further clarification regarding the healthy control data utilized in our study, sourced from the EZBioCloud platform. EZBioCloud is a comprehensive platform that hosts the taxonomic hierarchy of Bacteria and Archaea, represented by quality-controlled 16S rRNA gene and genome sequences. (doi:10.1099/ijsem.0.001755.) It also maintains and manages a significant collection of healthy control data, including datasets specifically representing the Korean population. We acknowledge the limitation associated with not using healthy controls recruited from the same institution and during the same period as our patients. This discrepancy could introduce variables that are not accounted for in our analysis, potentially influencing the comparability between the patient and control groups. We have recognized this limitation within our study and have discussed it explicitly in the limitations section of our manuscript.

DNA extraction and sequencing

- The reference provided (reference 34) does not seem to contain any information about the fecal collection kit (SMF-1) or the manufacturer's instructions. If this fecal collection kit is not a widely used or commercially available product, the authors should consider providing additional details about its provenance, validation, and suitability for gut microbiome studies.

Response: Thank you for your attention to the details regarding the fecal collection kit. In response to your valuable feedback, we have updated our manuscript to include detailed information about the buffer contained within the kit. Additionally, we have elaborated on the sample collection method employed in our study.

The added content is as follows: Fecal samples were collected using a fecal collection kit (SMF-1; ChunLab Inc., Seoul, Korea) that contained a lysis buffer (SDS 4%, Tris–HCL 50 mM, EDTA 50 mM, NaCl 500 mM) [36,37]. Patients were instructed to take a sample of their first faeces in the morning and place it in the kit. Samples were stored frozen until arrival at the laboratory, which took less than 3 days.

- How did the authors assess the qualities and quantities of extracted DNA.

Response: Samples were shipped to CJ Bioscience, and DNA was extracted for 16S rRNA analysis. Detailed information on the extraction protocol, including the specific steps and reagents used, is available in the the CleanPCR User Manual available at https://proteigene.com/app/uploads/2023/11/CleanPCR-User-Manual-v2.01.pdf.

For the DNA preparation and quality control (QC) methods employed in our study, we utilized the Promega Maxwell® RSC PureFood GMO and Authentication Kit for DNA extraction. To assess the quantity of the extracted DNA, we employed the BioTek Epoch™ Spectrometer. The condition of the extracted DNA was evaluated using agarose gel electrophoresis.

- Please provide the manufacturer’s name, city, country of the CleanPCR. This also valid for all reagents used.

Response: In response to your request for detailed information on the manufacturers of the reagents used in our study, we have added the following details to our manuscript: (Promega, Madison, WI, USA), (CleanNA, Waddinxveen, Netherlands),(Illumina, San Diego, CA, USA).

Statistical and bioinformatic analysis

- The authors are requested to provide more details about the data processing such as the database used to annotate the raw sequences and the threshold of identity percentage. I`m not sure if the authors made their sequences publicly available or not? If not, they need to submit their sequences and give their deposit number or doi.

Response: Thank you for your insightful comments and suggestions. We fully acknowledge the importance of making our data publicly available for the sake of transparency and reproducibility in scientific research. We had intended to comply with these standards by submitting our raw data to the NCBI database. Unfortunately, we have encountered an unforeseen and regrettable issue with our data storage server, which has resulted in the loss of access to the raw data necessary for submission. We understand the significance of this shortfall and deeply regret our inability to provide these crucial pieces of information. Please be assured that we have taken this matter very seriously and are currently implementing measures to prevent such occurrences in future research endeavors. We will ensure that all raw data is promptly submitted to relevant databases as part of our research protocol moving forward.

Thank you once again for your understanding and for bringing this important issue to our attention.

- Why LDA score ≥ 2 was used to estimate the effect size of each taxon?

Response: Thank you for your insightful comment. Upon reviewing your feedback and re-examining our data analysis process, we recognized an oversight in our initial selection of the LDA score threshold. Initially, we screened the data using an LDA score threshold of ≥2, based on the methodology described in a previous study (doi: 10.1186/gb-2011-12-6-r60). However, for our analysis, taxa with an LDA score of ≥3 were considered statistically significant. This discrepancy was an error on our part, and we have since corrected the threshold value to ≥3 in our manuscript to accurately reflect our analysis criteria. We are grateful for your attention to detail, which has helped us improve the accuracy and clarity of our work.

- The authors may need to explore their data and use the Phylogenetic Investigation of Communities by Reconstruction of Unobserved States (PICRUSt) approach to analyze both the relative abundances of the 16S rRNA genes and the predicted metabolic pathway.

Response: Thank you very much for your constructive suggestion with PICRUSt approach. While we recognize the value and relevance of incorporating the PICRUSt analysis,we regret to inform you that due to the current stage of our study and the limitations in our dataset, we are unable to implement this approach in the present work. We will apply this methodology in future studies. Thank you once again for your valuable feedback.

Results

Authors should provide information on the number of reads, chimeric sequences, valid reads, and read coverage.

Response: Thank you for your request for additional details regarding the sequencing data. In response to your query, we have included comprehensive information on the number of reads, identification of chimeric sequences, and read coverage within the manuscript as follows:

We obtained an average of 47,262 valid reads per sample (range: 26,417–85,273) and identified an average of 3,441 chimeric amplicons per sample (range: 450–12,568). The average read coverage achieved across all samples was 96.27%, with an average of 310 species identified per sample.

-The low resolution and lack of clear differentiation between the colors representing the bacterial genera make it difficult to discern the distinct differences of the core microbiota between the AA and HC groups.

Response: Thank you for your comment. We understand your concern about the difficulty in discerning the distinct differences due to the low resolution and the lack of clear color differentiation. Due to technical limitations, we encountered challenges in altering the colors within the bar graphs to more distinctly differentiate between the bacterial genera. However, in response to your feedback, we have taken steps to improve the overall clarity of the figure by enhancing the resolution of the entire image.

-Figure 1. What the ETC stand for?

Response: ETC stands for 'etcetera,' indicating items that are less than 1% in abundance. I'll spell out the abbreviation for clarity.

- Line 74-76: Figure 1 also shows that Prevotella accounts for more than 15% of the gut microbiome composition in the HC group, which is a significant proportion. Additionally, the prevalence of Prevotella in the AA group is almost comparable to that of Faecalibacterium, another genus identified by the authors as a core member. Therefore, I would suggest that the authors consider including Prevotella as an additional major genus, particularly given its substantial representation in both the AA and HC groups.

Response: Thank you for your insightful comment. Our intention was to focus on three major genera based on their abundance and relevance to our study's objectives. However, upon reviewing your suggestion and re-evaluating the data, we agree that Prevotella's significant presence in both groups warrants its inclusion as an additional major genus in our analysis. We have accordingly revised the manuscript to reflect Prevotella as a major genus alongside the others previously identified

- The authors should specify that the limited AA group consisted of 11 patients with a SALT grade of S1, and the moderate-to-severe AA group consisted of 8 patients with a SALT grade of S2 to S5.

Response: Thank you for your comment. we have made the modifications according to your suggestion.

- Again, the authors should consider increasing the resolution of the figures presented in the manuscript. Higher resolution figures would improve the clarity and visual quality of the data visualization, making it easier for readers to interpret the information conveyed in the images.

Response: Thank you for your valuable comment. We appreciate your suggestion and have replaced the previous images with higher resolution figures to enhance clarity and visual quality, making it easier for readers to interpret the data presented. Thank you again for helping us improve our manuscript.

Discussion:

- The claim about the number of bacterial species in the human gastrointestinal tract being 500-1,000 originated from the review article by Jian Xu and Jeffrey I. Gordon (PMID 12923294), and not from the reference cited by the authors.

Response: Thank you for pointing out the correct source for the claim regarding the number of bacterial species in the human gastrointestinal tract. We have updated the reference accordingly.

- Line 138-143. The authors should reconcile the apparent contradiction in their statements regarding the existing studies on the role of gut microbiota in the pathogenesis of AA. On one hand, they state that "several studies have identified a potential role for gut microbiota in the pathogenesis of AA", suggesting a substantial body of research on this topic. However, they then go on to say that "studies characterizing the gut microbiota in AA patients have been limited".

Response: We agree with your comment and have revised the some expressions to clarify our position on the current state of research in this area. Thank you for helping us improve the accuracy and consistency of our manuscript.

- Line 154-161. The authors need to provide potential explanations for the observed differences. The authors acknowledge that the findings of Moreno-Arrones et al. and Rangu et al. may have been affected by the study populations (alopecia universalis patients and children with AA), but they do not explore this aspect in more detail. It would be valuable to discuss potential factors that could contribute to the varying results reported in the literature.

Response: In response to the valuable feedback regarding the exploration of factors contributing to the varying results reported in the literature, we have expanded our discussion to consider potential variables that may influence the observed outcomes.

- Line 161 correct “univeralis” to “universalis”

Response: Thank you for pointing out the typo. We will correct "univeralis" to "universalis"

- Lines 162-168. The authors have noted the increased incidence of these genera in AA patients, but they do not provide mechanistic insights into how these specific bacteriomes may contribute to the pathophysiology of AA. This is also true for the F/B ratio and the less dominant family Ruminococcaceae

Response: Thank you for your insightful comment. We acknowledge this as a limitation of our current research. Due to the scope and resources available for this study, we were unable to delve into the mechanistic pathways that these bacterial genera may influence in the context of AA. We will apply this methodology in future studies. Thank you once again for your valuable feedback.

- Lines 180-181 provide an adequate reference (e.g PMID: 32438689)

Response: Thank you for pointing that out. We have updated the reference as suggested

- Line 178-185. What is the potential clinical implications of this observation?

Response: Thank you for your comment. Following your suggestion, we have added the following statement to our manuscript: The elevated F/B ratio observed in our study among AA patients further suggests that dysbiosis may play a role in the pathogenesis of AA, underscoring the potential for mi-crobial balance as a target for therapeutic intervention.

- Some editing for English language is required throughout the manuscript

Response: Thank you for pointing out the necessity for editing. Our manuscript underwent a professional proofreading service before submission, despite this precaution, it appears that some errors remained unaddressed. We conducted an additional thorough review of the manuscript ourselves.

We are grateful for your guidance in enhancing the quality of our work. Your insights have significantly contributed to enhancing the quality and clarity of our study.

Reviewer 2 Report

Comments and Suggestions for Authors

Major comments:

1)      How authors ensured that the collected sample is not contaminated with environmental bacteria/microorganisms? Is there any special instruction or precaution followed by patients? Timing of stool collection not mentioned

2)      Explain the rationale of using  ‘region from V3 to V4 of 16S rRNA gene was targeted and amplified with fusion primers 341F and 805R.’(line#226). (Authors performed the phylogenetic unit analysis? (using Full-length 16S ribosomal RNA (rRNA) gene sequence)?  

3)      It is recommended to include the heat map showing the relative abundance of the bacterial species between the two groups.

4)      Demographic detail of patient and study group is poorly described. Is there any difference observed in bacterial species diversity and/or relative abundance with respect to age and gender of the patients.

5)      Is it possible to determine the Effect size estimates of gene ortholog abundance/species diversity/abundance with age and disease severity?

6)      Many recent papers were not cited and discussed? (e.g. DOI: https://doi.org/10.5114/ada.2022.120453).

7)      Past history of medication in patient must be included, how authors can ensure that previous medication has not altered the gut microbiota of the patients. Further,  gut microbiota is affected by various factors like diet, nutrition, age, gender, lifestyle etc., it is very difficult to relate the findings to disease etiology considering the limited sample size. Authors must add appropriate references and discuss the same to address the effect of above mentioned factors.

Minor comments:

1)      Figure labels were poorly presented, revise and increase the fonts of the labels

2)      Proofreading of manuscript is required for ‘typos’

Comments on the Quality of English Language

Manuscript require proofreading for minor corrections.

Author Response

First and foremost, we would like to express our sincere gratitude for your thorough review and valuable comments on our manuscript. Your insights have significantly contributed to enhancing the quality and clarity of our study. We have carefully considered your suggestions and have made the following revisions to our manuscript to address the concerns raised.

Major comments:

1)      How authors ensured that the collected sample is not contaminated with environmental bacteria/microorganisms? Is there any special instruction or precaution followed by patients? Timing of stool collection not mentioned

Response: Thank you for your valuable comment. In response to your concerns, we have added detailed information about the timing and method of fecal sample collection to our manuscript.

The relevant additions are as follows: Fecal samples were collected using a fecal collection kit (SMF-1; ChunLab Inc., Seoul, Korea) that contained a lysis buffer (SDS 4%, Tris–HCL 50 mM, EDTA 50 mM, NaCl 500 mM) [36,37]. Patients were instructed to take a sample of their first faeces in the morning and place it in the kit. Samples were stored frozen until arrival at the laboratory, which took less than 3 days.

2)      Explain the rationale of using ‘region from V3 to V4 of 16S rRNA gene was targeted and amplified with fusion primers 341F and 805R.’(line#226). (Authors performed the phylogenetic unit analysis? (using Full-length 16S ribosomal RNA (rRNA) gene sequence)?  

Response: References for using primers are as follows; Fadrosh et al. An improved dual-indexing approach for multiplexed 16S rRNA gene sequencing on the Illumina MiSeq platform. Microbiome 2014 2:6

3)      It is recommended to include the heat map showing the relative abundance of the bacterial species between the two groups.

Response: Thank you for your valuable comment. Following your recommendation, we have incorporated a heat map to illustrate the relative abundance of bacterial species between the two groups. This addition has indeed enriched our research by providing a clearer visual representation of the microbiome differences. We are grateful for your suggestion, which has significantly enhanced the quality of our study. Thank you for your valuable input.

4)      Demographic detail of patient and study group is poorly described. Is there any difference observed in bacterial species diversity and/or relative abundance with respect to age and gender of the patients.

Response: Thank you for your insightful observation regarding the demographic details of our patient and study group. We acknowledge that a more detailed analysis concerning the impact of age and gender on bacterial species diversity and relative abundance would significantly enhance the understanding of our findings. However, due to the limited number of participants in our current study, it was challenging to conduct a robust analysis segmented by these demographic factors. We fully agree that exploring these aspects in future studies with a larger cohort would be highly beneficial.

5)      Is it possible to determine the Effect size estimates of gene ortholog abundance/species diversity/abundance with age and disease severity?

Response: Thank you for your question. As previously mentioned, the limited number of participants in our study posed a challenge for conducting separate analyses based on age and disease severity. This limitation prevented us from performing the detailed investigation that your question entails. However, I wholeheartedly agree with your insightful suggestion and share your interest in exploring these relationships in future research.

6)      Many recent papers were not cited and discussed? (e.g. DOI:https://doi.org/10.5114/ada.2022.120453).

Response: Thank you for your comment regarding the citation and discussion of recent papers. Thanks to your guidance, we have added an additional reference and incorporated a summary of its contents into our manuscript. To date, a total of four studies have been published on the intestinal microflora in patients with alopecia areata. Your insightful feedback has enabled us to mention all four studies in our discussion, enriching our manuscript's context and comprehensiveness. We are grateful for your contribution, which has significantly enhanced the quality of our work.

7)      Past history of medication in patient must be included, how authors can ensure that previous medication has not altered the gut microbiota of the patients. Further,  gut microbiota is affected by various factors like diet, nutrition, age, gender, lifestyle etc., it is very difficult to relate the findings to disease etiology considering the limited sample size. Authors must add appropriate references and discuss the same to address the effect of above mentioned factors.

Response: Thank you for your insightful comments.We fully agree with the importance of considering these variables in our analysis. To mitigate the potential impact of previous medications on the gut microbiota, we carefully established exclusion criteria for participant recruitment. Specifically, we excluded individuals with a history of receiving oral or intravenous antibiotic treatment, systemic immunomodulators, extreme diets, or probiotics in the previous 12 weeks. This approach was intended to minimize the influence of recent external factors on the gut microbiota composition of our study participants.

We believe that these measures, along with our detailed participant selection process, help to address concerns regarding the potential confounding effects of past medication and other factors on our findings.

Minor comments:

1)      Figure labels were poorly presented, revise and increase the fonts of the labels

Response: Thank you for your valuable feedback. In response to your comments, we replaced all the current figures with high-resolution versions.

2)      Proofreading of manuscript is required for ‘typos’

Response: Thank you for pointing out the necessity for thorough proofreading to correct typos in our manuscript. Our manuscript underwent a professional proofreading service before submission, despite this precaution, it appears that some errors remained unaddressed. We conducted an additional thorough review of the manuscript ourselves.

We are grateful for your guidance in enhancing the quality of our work. Your insights have significantly contributed to enhancing the quality and clarity of our study.

Reviewer 3 Report

Comments and Suggestions for Authors

The authors investigatd the involvement of gut microbiota in alopecia areata (AA) and found significant differences in bacterial community composition between the AA group and healthy controls, suggesting the gut microbiota's potential role in AA. The research presents both interesting and significant findings.

1.In the Introduction, the authors insufficiently justify the investigation of the relationship between gut microbiota and alopecia areata (AA). The statement, “Several independent lines of evidence support the idea that gut microbiota can contribute to the pathogenesis of AA,” lacks citation of relevant references. To strengthen this section, it is necessary to include additional references demonstrating the immunological relevance of this research topic.

2.Given the multiple comparisons conducted, setting the threshold for a significant p-value at < 0.05 is inappropriate. It is essential to apply corrections for multiple comparisons, such as the Bonferroni correction, to ensure the statistical validity of the results.

Author Response

First and foremost, we would like to express our sincere gratitude for your thorough review and valuable comments on our manuscript. Your insights have significantly contributed to enhancing the quality and clarity of our study. We have carefully considered your suggestions and have made the following revisions to our manuscript to address the concerns raised.

1.In the Introduction, the authors insufficiently justify the investigation of the relationship between gut microbiota and alopecia areata (AA). The statement, “Several independent lines of evidence support the idea that gut microbiota can contribute to the pathogenesis of AA,” lacks citation of relevant references. To strengthen this section, it is necessary to include additional references demonstrating the immunological relevance of this research topic.

Response: Thank you for your insightful comments. In response to your feedback, we have added the following relevant references and related citations: Rebello and Xie observed hair regrowth in AA patients following fecal material trans-plantation for refractory bowel diseases [12,13]. Preclinical studies in murine models have indicated the possible role of diet and the gut microbiome in the onset of AA [14,15]. Recent clinical studies have characterized the gut microbiota of AA patients [16-19]

Furthermore, we would like to mention that our discussion section already incorporates additional lines of evidence from animal studies and recent research using patient data, which collectively support the hypothesis that alterations in gut microbiota could influence AA development.

We are grateful for your guidance in enhancing the quality of our work.

2.Given the multiple comparisons conducted, setting the threshold for a significant p-value at < 0.05 is inappropriate. It is essential to apply corrections for multiple comparisons, such as the Bonferroni correction, to ensure the statistical validity of the results.

Response: Thank you for your critical observation regarding the statistical approach. We understand your concern about the potential for type I errors when multiple comparisons are made without appropriate corrections. In our study, we utilized the Wilcoxon rank-sum test for the analysis. We acknowledge that the initial description of our statistical methodology and the presentation of p-values may have inadvertently led to confusion. We revised parts that may cause misunderstanding.

We are grateful for your guidance in enhancing the quality of our work. Your insights have significantly contributed to enhancing the quality and clarity of our study.

Reviewer 4 Report

Comments and Suggestions for Authors

The Communication paper entitled Exploring the Role of Gut Microbiota in Alopecia Areata: An Analysis of Gut Microbiota in Patients with Alopecia Areata approaches gut microbiota analysis in alopecia areata (AA) as a potential new source of biomarkers and therapeutic targets in AA and other autoimmune diseases.

The authors provide results and insights regarding gut microbial composition and diversity in AA patients with different severity degrees of the condition versus healthy controls (HC). Thus, authors data refers to critical bacteria of gut microbiota composition connected to AA compared to HC at the genus level. Findings show that Firmicutes to Bacteriodetes ratio was significantly higher in AA group compared with the HC group, suggesting bacterial dysbiosis. In addition, regarding the α-diversity of gut microbiota the authors find that there are no statistically significant differences between experimental groups according to ACE, Chao1, Shannon and Simpson parameters.

The authors suggest bacterial biomarkers for AA using LDA, LEfSe analysis and they found an enriched occurrence of Blautia, Dorea, Collinsella, Anaerostipes and Eubacterium_g5 genera compared with HC, with the mention that the specific role of these genera in different autoimmune conditions needs further investigation.

The manuscript provides a clear picture of microbiome analysis as a new tool for the discovery of biomarkers and therapeutic targets in AA, with application to other autoimmune conditions.

Some suggestions for improving the manuscript:

- All figures must be done in a higher resolution because they are difficult to analyze;

- At Section "4.1 Participants" the authors state that “Age- and sex-matched HC data were extracted at random from a cohort of men and women in Korea using the EZBioCloud public database [33]. At Section "4.3. Statistical and bioinformatic analysis" it is mentioned that “The microbiome taxonomic profiling cloud of EZBiocloud was used to analyze the taxonomic bacterial profiling [35]. Is EZBioCloud the same database used for HC data extraction and microbiome profiling? Please check references 33 and 35.

- I would suggest to revise the paper title to manage the word repetition ("Alopecia Areata")

Author Response

First and foremost, we would like to express our sincere gratitude for your thorough review and valuable comments on our manuscript. Your insights have significantly contributed to enhancing the quality and clarity of our study. We have carefully considered your suggestions and have made the following revisions to our manuscript to address the concerns raised.

- All figures must be done in a higher resolution because they are difficult to analyze;

Response: Thank you for your valuable feedback regarding the resolution of the figures included in our manuscript. In response to your comments, we replaced all the current figures with high-resolution versions.

- At Section "4.1 Participants" the authors state that “Age- and sex-matched HC data were extracted at random from a cohort of men and women in Korea using the EZBioCloud public database [33]”. At Section "4.3. Statistical and bioinformatic analysis" it is mentioned that “The microbiome taxonomic profiling cloud of EZBiocloud was used to analyze the taxonomic bacterial profiling [35]”. Is EZBioCloud the same database used for HC data extraction and microbiome profiling? Please check references 33 and 35.

Response: Thank you for your inquiry regarding the use of the EZBioCloud database in our study, specifically in Sections "4.1 Participants" and "4.3. Statistical and bioinformatic analysis".

Yes, EZBioCloud is the same database used for both the extraction of age- and sex-matched healthy control (HC) data and the microbiome profiling in our study. EZBioCloud is a comprehensive platform that hosts the taxonomic hierarchy of Bacteria and Archaea, represented by quality-controlled 16S rRNA gene and genome sequences. (doi:10.1099/ijsem.0.001755.) Furthermore, EZBioCloud maintains and manages a significant collection of healthy control data, including datasets specifically representing the Korean population. This aspect of EZBioCloud enabled us to extract relevant HC data for our study. We hope this response satisfactorily addresses your query.

- I would suggest to revise the paper title to manage the word repetition ("Alopecia Areata")

Response: Thank you for your constructive feedback regarding the repetition of "Alopecia Areata" in our paper's title. After careful consideration, we have decided to revise the title to "Exploring the Role of Gut Microbiota in Patients with Alopecia Areata". We hope that this revision meets your approval, and we are grateful for your guidance in improving our manuscript.

We are grateful for your guidance in enhancing the quality of our work. Your insights have significantly contributed to enhancing the quality and clarity of our study.

Round 2

Reviewer 1 Report

Comments and Suggestions for Authors

The authors have largely addressed the questions I raised in my previous review. I recommend proceeding with the publication of this article.